# Trade-Off Regulation in Plant Growth and Stress Responses Through the Role of Heterotrimeric G Protein Signaling

**DOI:** 10.3390/plants13223239

**Published:** 2024-11-19

**Authors:** Horim Lee

**Affiliations:** Department of Biotechnology, Duksung Women’s University, Seoul 01369, Republic of Korea; hrlee1375@duksung.ac.kr; Tel.: +82-2-901-8753

**Keywords:** heterotrimeric G protein, developmental plasticity, trade-off, plant immunity, thermomorphogenesis

## Abstract

Unlike animals, plants are sessile organisms that cannot migrate to more favorable conditions and must constantly adapt to a variety of biotic and abiotic stresses. Therefore, plants exhibit developmental plasticity to cope, which is probably based on the underlying trade-off mechanism that allocates energy expenditure between growth and stress responses to achieve appropriate growth and development under different environmental conditions. Plant heterotrimeric G protein signaling plays a crucial role in the trade-off involved in the regulation of normal growth and stress adaptation. This review examines the composition and signaling processes of heterotrimeric G proteins in plants, detailing how they balance growth and adaptive responses in plant immunity and thermomorphogenesis through recent advances in the field. Understanding the trade-offs associated with plant G protein signaling will have significant implications for agricultural innovation, particularly in the development of crops with improved resilience and minimal growth penalties under environmental stress.

## 1. Introduction

Unlike animals, plants cannot actively select optimal environmental conditions for growth and development. Therefore, plants must exist in constant interactions with a wide range of biotic and abiotic stresses that they are exposed to in the confined place they live throughout their lifespan. Thus, the impact of adverse stress environments on plant growth is an important scientific problem for an integrated understanding of fundamental growth and development under ever-changing environmental conditions. In addition, the impact of adverse stresses on plant growth is a critical issue for agriculture and food security, particularly in response to the extreme environmental challenges expected in the future (e.g., climate change) [1]. Plants can enhance certain traits during growth and development while restricting other traits. Hence, plants under stress conditions exhibit developmental plasticity [2], which is associated with a very active allocation of energy use between growth and stress response (Figure 1). Under optimal growth conditions, plants normally suppress the stress response programs, including abscisic acid (ABA)-mediated morphogenesis and gene expression, which is necessary for resistance or adaptation to biotic and abiotic stresses to avoid unnecessary energy expenditure for normal growth and development [3]. In contrast, plants under stress conditions relatively reduce the energy expenditure required for normal growth and development to use energy for adaptation against various stresses. These plastic growth phenomena are generally observed in the active growth retardation associated with stress tolerance based on the energy allocation in plants exposed to stress environments [4] or in an overexpression of stress-responsive genes in transgenic plants [5,6]. Therefore, stress-induced developmental plasticity is likely linked to the underlying trade-off mechanism that selectively allocates the available energy/nutrient resources to plants in response to environmental changes.

Heterotrimeric G proteins (referred to as G proteins), a complex comprising Gα, Gβ, and Gγ, are an important signaling system that interacts with G protein-coupled receptors (GPCRs) to sense a variety of stimuli, such as nutrients and hormones, and contributes to the regulation of the physiological balance between healthy and stressed conditions in animals [7]. In contrast, although heterotrimeric G protein complexes also exist in plants, their simpler composition and lack of GPCR suggest that plant G protein signaling mechanisms are likely to be regulated differently from those in animals [8,9]. Nevertheless, plant G protein signaling has been shown to be similarly involved in the physiological responses to external stimuli, such as biotic and abiotic stresses, as well as in basic growth and development based on studies using G protein component mutants, such as *gpa1* (loss of function of the Gα subunit), *agb1* (l-o-f of the Gβ subunit), and *agg1/2/3* (l-o-f of the Gγ subunits) [8,10]. Recent findings have revealed alleviated growth inhibition under biotic stress and thermomorphogenesis induced by high ambient temperature in l-o-f *Arabidopsis* G protein component mutants [11], suggesting the significant role of G protein signaling in regulating the trade-off between growth and stress response for developmental plasticity. Fundamentally, G protein-mediated growth inhibition under stress conditions is ultimately associated with the underlying cellular behavior, including cell proliferation and death [12,13]. This cell survival mechanism under nutrient starvation conditions is also regulated by the key trade-off modulator target of rapamycin (TOR) kinase [14]. Here, this review provides an overview of the molecular processes and roles of G protein signaling in plant growth, development, and stress responses. In addition, this review discusses the recent advances in understanding the function of G protein signaling in the trade-off associated with plant immunity and thermomorphogenesis and provides a perspective potentially in connection to trade-off modulators.

## 2. Heterotrimeric G Protein Signaling Acts to Balance Growth and Stress Responses in Plants

### 2.1. Heterotrimeric G Protein Subunits in Plants

Compared to vast numbers of G protein subunits in animals [15], most plants show relatively simple compositions of canonical G protein signaling components. For example, *Arabidopsis* has one Gα, one Gβ, and three Gγ subunits [16,17,18,19,20], and rice has one Gα, one Gβ, and five Gγ subunits [21]. Although several Gγ subunits are found in plants, the canonical Gγ prototype that contains the isoprenylation motif at the C-terminus to anchor the plasma membrane in animals is revealed as two *AGG1* and *AGG2* in *Arabidopsis* and one *RGG1* in rice [9,21,22]. Other types of Gγ subunits lack the isoprenylation motif or have extended C-terminal domains with highly enriched cysteine residues [21]. In addition, most plants have non-canonical plant-specific Gα subunits such as the extra-large GTP-binding protein (XLG), which contain the C-terminal domains homologous with canonical Gα subunits and the extensive N-terminal region including a nuclear localization signal, a nuclear export signal, and a cysteine-rich sequence [8,23,24]. The l-o-f triple mutants for all three genes, *XLG1*, *XLG2*, and *XLG3*, encoded in the *Arabidopsis* genome showed similar phenotypes for ABA/sugar sensitivity, defense response, and root-wave response like those in the *agb1* mutant, indicating the associated roles of XLGs with canonical G protein signaling [25,26]. Although mammals have XLGs in addition to five major types of Gα subunits, including Gα_s_, Gα_i_, Gα_q/11_, Gα_12/13_, and Gα_v_ [27], they are produced by alternative splicing from canonical Gα genes unlike unique genes in plants [23].

Despite the additional non-canonical Gα and Gγ subunits, plants have a relatively limited diversity of combinations to form ternary complexes compared to those in animals, which contain almost 40 Gαβγ components [15]. Nevertheless, G protein signaling in plants affects diverse biological processes ranging from fundamental growth and development to adaptive responses to biotic and abiotic stresses [8,9,28]. Thus, the ability to perform multiple functions with such a low level of combinatorial complexity suggests that plant G protein signaling may be involved in simple and common processes that regulate growth under optimal or stressful conditions. Although the stresses are diverse, the phenomenon resulting from trade-off regulation is commonly found as growth inhibition for proper growth and development under adverse stress conditions.

### 2.2. Molecular Processes of Heterotrimeric G Protein Signaling in Plants

Most GDP-bound Gαβγ trimeric complexes are activated by GPCRs sensing exogenous signal ligands through the conventional mechanism of G protein signaling in animals [29]. The intrinsic exchange rate from the inactive GDP-bound Gα subunit to the active GTP-bound Gα subunit is very slow. Therefore, GPCRs acting as guanosine exchange factors (GEFs) are required to induce G protein signaling, allowing the trimeric complexes to separate into active Gα and active Gβγ to interact with the downstream effectors, e.g., adenylyl cyclase and ion channels, respectively. Once G protein signaling is activated, the deactivation process occurs through the spontaneous intrinsic GTP hydrolysis of the Gα subunit or GTPase-accelerating proteins (GAPs), also known as a regulator of G protein signaling (RGS) proteins, to terminate G protein signaling [29,30]. In contrast to animals, activation of the Gα subunit is not a rate-limiting step in plants because the GDP from the Gα subunit is released and exchanged spontaneously with GTP without the effort of GEFs [31,32]. Interestingly, the exchange rate of the plant Gα subunit is similar to that of the constitutively active version of the animal Gα mutant subunit [33]. Furthermore, the GPCR that activates the Gα subunit in animals has not been identified in plants [32], suggesting that plant G protein signaling is probably self-activated in a GPCR-independent manner. Instead, the deactivation process is the rate-limiting step in plant G protein signaling in contrast to animals because the Gα subunit has an intrinsically slow activity of GTP hydrolysis [32,34]. Therefore, the function of RGS proteins as GAPs to promote GTP hydrolysis is essential for deactivating the active plant Gα subunits. Another distinctive feature of plant RGS proteins acting as a GTPase is that they contain a seven-transmembrane (7TM) domain in the N-terminal region, similar to the animal metabotropic glutamate GPCR subfamily [34,35,36]. The catalytic domain of RGS is located in the cytoplasmic C-terminal region, which is homologous to the animal RGS protein GAP [34,35]. In the active/inactive cycle of G protein signaling, the critical rate-limiting step differs according to the intrinsic catalytic properties and is critically regulated by GPCRs and RGS proteins in animals and plants, respectively, to overcome the impaired catalytic activities.

The purified C-terminal RGS-box domain in *Arabidopsis* 7TM-RGS protein, known as AtRGS1, has been reported to be able to accelerate the GTPase activity of GPA1 in vitro [34,35,37]. In addition, genetic analysis showed that the *gpa1* mutant has a shorter hypocotyl length, whereas *atrga1* and constitutively active *GPA1* increased hypocotyls in the dark [35]. Hence, AtRGS1 negatively regulates the activity of GTP-bound GPA1, which is involved in cell elongation and proliferation. Moreover, the 7TM AtRGS1 protein is a putative receptor for glucose ligands [35,38,39]. Several findings showed that glucose triggers the endocytosis of its receptor AtRGS1 from the plasma membrane to activate Gα signaling through physical uncoupling between GPA1 and its inhibitor AtRGS1 [34,40,41]. The effects of glucose on G protein signaling are supported by genetic analyses. For example, the seedling growth arrest induced by high glucose concentrations was alleviated in the *atrgs1* mutant. In addition, hypersensitive growth arrest was exhibited in *gpa1* and *atrgs1 gpa1* mutant seedlings or transgenic plants overexpressing *AtRGS1* [34], suggesting that *AtRGS1* and *GPA1* are involved in glucose signaling in the same genetic pathway for plant growth and that G protein signaling is vital for sugar signaling.

### 2.3. Heterotrimeric G Protein Signaling in Plant Immunity

Under biotic stress conditions, plants directly respond to pathogen attacks (e.g., bacteria, fungi, and oomycetes) via microbe-associated molecular pattern (MAMP)-triggered immunity (MTI) as an innate immune system [42,43]. Pathogen signals, MAMPs, are recognized by the pattern recognition receptors (PRRs) on the cell surface [42]. One of the best-characterized plant MAMP signals is the flg22 peptide derived from bacterial flagellin, which is recognized by the PRR flagellin-insensitive 2 (FLS2) receptor, a member of the receptor-like kinase (RLK) family (Figure 2) [42]. Upon the perception of flg22, FLS2 forms a heterodimer complex with the co-receptor BRI1-associated kinase 1 (BAK1) to initiate the downstream immune responses, including mitogen-activated protein kinase (MAPK) activation, reactive oxygen species (ROS) production, and immune gene expression [42,44]. Lu et al. [45] reported that botrytis-induced kinase 1 (BIK1), which encodes the receptor-like cytoplasmic kinase (RLCK), plays an important role in transducing intracellular signals from the flg22-mediated FLS2-BAK1 receptor complex to downstream responses. Accordingly, BIK1 was phosphorylated rapidly by flg22 via BAK1 and activated BIK1 reciprocally phosphorylated FLS2 and BAK1 for positive propagation of the flg22 signaling pathway [45].

Plant G protein signaling is also known to be involved in innate immunity. Liang et al. [46] reported that G protein complexes, including XLG2 and AGB1 with AGG1 or AGG2, are required for FLS2-mediated immunity. They showed that non-canonical Gα XLG2, but not canonical Gα GPA1, is critical for the flg22-mediated response through direct interactions with the FLS2 and BIK1 complex [46]. In l-o-f *xlg2* mutant leaves, increased *Pseudomonas syringae* pv *tomato* (*Pst*) DC3000 growth and reduced ROS production were observed under flg22-infiltrated conditions compared to the water-infiltrated control [46], suggesting the resistance function of *XLG2* against *Pst* DC3000 via flg22-mediated ROS production. After the perception of flg22, XLG2 dissociated from AGB1 was phosphorylated at the N-terminus by activated BIK1, and phosphorylated XLG2 enhanced the activity of NADPH oxidase RbohD, which produces ROS [46]. In addition, BIK1 activated by flg22 also phosphorylated RGS1 at Ser428 and Ser431 to activate the GTP-bound Gα subunit through the dissociation from the FLS2–G protein complex [47]. Previous studies have shown that *AGB1*, *AGG1*, and *AGG2* are involved in pathogen resistance and ROS production [48,49], suggesting that the XLG2-AGB1-AGG1/2 G protein signaling module is required for FLS2-mediated immunity.

Previous studies have shown that the canonical Gα GPA1 is not essential for flg22-induced ROS production and resistance, as the *gpa1* mutant showed similar susceptibility to different pathogenic *P. syringae* strains compared to the wild type (WT) [48,49]. Although GPA1 does not appear to be involved in FLS2-mediated basal immunity based on these observations, it has been shown to play an important role in FLS2-mediated stomatal resistance [50]. Stomatal opening is critical for pathogen entry [51], and GPA1 is required for stomatal closure due to flg22 treatment [52]. Although the susceptibility of *Pst* DC300 was not indistinguishable between WT and *gpa1* [48,49], the *gpa1* mutant showed increased susceptibility, similar to that of the *fls2* mutant, to the COR-deficient *Pst* DC3000 mutants, which have a defect in stomatal reopening [50]. Moreover, GPA1 and AGG1/2 interact with another defense-related RLK, chitin elicitor receptor kinase 1 (CERK1), but not FLS2, through yeast split ubiquitin and bimolecular fluorescence complementation (BiFC) assays [53]. These reports suggest that GPA1 is likely involved in plant immunity through a distinct signaling module with Gβγ subunits to different RLKs.

Interestingly, recent studies reported that the positive function of the conventional Gα GPA1 instead of XLG2 mediates the responses of flg22 in plant immunity through different experimental conditions. Xue et al. [54] reported that the bacterial growth of the less virulent *P. syringae* pv. *Maculicola* ES4326, but not *Pst* DC3000, was higher in the flg22-treated *gpa1* mutants than in the WT, indicating compromised flg22-mediated immunity in *gpa1*. In addition, the *gpa1* mutant showed slightly lower ROS production upon flg22 treatment than the WT. Similarly, enhanced bacterial growth and susceptibility in *gpa1* as well as *agb1* and *agg1 agg2* in response to diverse host and nonhost *Pseudomonas* pathogens has been reported [55]. Moreover, an increase in the overall phosphorylation in GPA1 was also induced by flg22, and GPA1 phosphorylation was abolished in the *bak1* null mutant [54,56]. These results suggest that flg22 triggers the downstream immune responses, including GPA1, via BAK1, which probably acts as a kinase for GPA1. Additional in vitro analysis revealed RGS1 phosphorylation upon flg22 perception through defense-related RLKs, including BAK1. This phosphorylated RGS1 induced the dissociation of the FLS2-BAK1-GPA1-AGB1 complex to activate G protein signaling [54,56,57], proposing a similar role of RLKs in plants to that of GPCRs in animals to trigger the self-activation of the Gα subunit. Overall, these data suggest that complex and diverse G protein signaling modules, including GPA1 or XLGs, are involved in plant immunity.

### 2.4. Trade-Off Regulation Between Plant Growth and Defense via Heterotrimeric G Protein Signaling

In addition to the MAMP-mediated early responses (within 30 min), including ion fluxes, oxidative burst, MAPK activation, receptor endocytosis, and gene expression, plants also have late responses (hours to days), including seedling growth inhibition [42]. The inhibition of plant growth by MAPMs, such as flg22, usually appears to be the result of a trade-off to modulate energy use from normal growth to enhance pathogen resistance [58]. Recently, the *agb1* mutant was almost insensitive to growth inhibition under biotic stress conditions but not *gpa1* [11]. Yang et al. [11] reported that an flg22 treatment severely reduced the primary root length of WT and *gpa1* seedlings. In contrast, the root growth inhibsition of *agb1* single- and *agb1 gpa1* double-mutant seedlings was almost abolished. Consistent with previous studies [49,54,59], early responses, such as MAPK activation and immune gene expression, induced by flg22 were unaffected in *gpa1* and *agb1* mutants compared to WT [11]. Therefore, the flg22-mediated early and late responses are likely to be uncoupled processes. Moreover, G protein signaling is likely to be important for growth regulation associated with plant immunity by modulating energy use. On the other hand, no growth inhibition was observed in *gpa1* compared to *agb1* [11]. Therefore, two possible scenarios for these results can be inferred based on previous reports. First, activated Gβγ complexes, including AGB1, that dissociate from the inactive form of the heterotrimeric complex after flg22 induction may function primarily in growth regulation related to the immune response. Second, the atypical Gα XLG2-mediated G protein signaling module may be important for growth regulation instead of GPA1.

The *Arabidopsis* Gα and Gβ subunits are also involved in regulating shoot apical meristem (SAM) development through the CLAVATA (CLV)-WUSCHEL (WUS) signaling pathway [60,61]. For example, *agb1* was isolated from suppressor mutant screens using *clv2* because it exhibited an enhanced phenotype of an enlarged *clv2* SAM size, and AGB1 controlled SAM maintenance through protein–protein interactions with RECEPTOR-LIKE PROTEIN KINASE 2 (RPK2) [60]. Similarly, the maize Gα subunit COMPACT PLANT2 (CT2) was also reported to interact with the CLV2 ortholog FASCIATED EAR 2 (FEA2) to control SAM size [62]. Recently, maize *XLGs* and *ZmGB1*, which encode atypical Gα and Gβ subunits in maize, respectively, play important roles in SAM development according to CRISPR-Cas9 analysis [63,64]. Unlike *Arabidopsis*, the null mutant of *ZmGB1* showed a seedling-lethal phenotype similar to that of rice [65], suggesting that the monocot Gβ subunit is crucial for growth and survival. Interestingly, Wu et al. [64] reported that the lethal phenotype of *ZmGB1* was caused by an autoimmune response but not by growth arrest. They showed high levels of trypan blue and DAB accumulation, which indicate cell death and H_2_O_2_ production, respectively. Moreover, immune marker genes, such as *PATHOGENESIS-RELATED PROTEIN* (*PR1*) and *PR5*, are strongly expressed in the CRISPR/Cas9-mediated knockout mutant of *ZmGB1* (*Zmgb1^CR^*), suggesting a correlation between seedling death and autoimmunity. In addition, through suppressor analysis by crossing *Zmgb1^CR^* with a tropical maize line CML103, they reported that suppressed mutants derived from *Zmgb1^CR^* exhibited reduced *PR* gene expression and an enlarged SAM size [64]. These data suggest that the trade-off regulation between growth and immunity depends on the activity of the Gβ subunit.

### 2.5. Trade-Off Regulation by Heterotrimeric G Protein Signaling in Thermomorphogenesis

Temperature is one of the most critical abiotic and environmental factors for plastic growth and development in plants. Compared to the normal growth temperature (22–23 °C) of *Arabidopsis*, the high temperature affecting growth and development can mainly be divided into two types [66]. The first type is extremely high temperatures, which are recognized as heat stress (>40 °C), which may cause immediate cell death [67]. *Arabidopsis* plants can sometimes acquire thermotolerance by being exposed to moderately high temperatures (<37 °C) as a heat acclimation process [67,68]. Heat stress suppresses plant growth and development, including seed germination, seedling growth, and pollination. Under this pressure, plants have developed evolutionarily to acquire adaptive priming mechanisms for thermotolerance responses [69], in which the trade-off modulating the re-distribution of energy allocation between growth and stress responses is regulated by heat shock transcription factors (HSFs) and plant hormones, such as gibberellins, brassinosteroids, ABA, and salicylic acid [70]. In addition, the function of TOR has been reported to act as a modulator of the trade-off between growth and heat stress [71]. Sharma et al. [71] reported that glucose–TOR signaling plays a vital role in the adaptation to heat stress responses by reprograming the expression profiles and epigenetic regulation. As a result, the seedlings overexpressing *TOR* showed a significantly enhanced growth phenotype, whereas the *tor* mutant showed a sensitive phenotype.

The second type of high temperature is associated with warmer and relatively non-stressful conditions, known as high ambient temperatures (27–32 °C), which can also affect the morphological and developmental changes known as thermomorphogenesis, including the inhibition of seed germination, enhanced hypocotyl/petiole elongation, and induced leaf thermonasty and early flowering at high ambient temperature, mainly through the PHYTOCHROM-INTERACTING FACTOR 4 (PIF4)-mediated pathways [66,72]. PIFs, including PIF4, are accumulated under high ambient temperature by inhibiting phytochrome B [73], accelerating the expression of auxin biosynthetic genes, such as *YUCCA8* (*YUC8*) and *TRYPTOPHAN AMINOTRANSFERASE OF ARABIDOPSIS1* (*TAA1*), increasing auxin levels and increasing cell elongation in hypocotyls [74,75]. Recently, the LONG HYPOCOTYL5 (HY5)-PIF signaling module was reported to function in the developmental trade-off to balance shoot and root growth at high ambient temperatures [76]. Overall, survival against or thermomorphogenic responses to a wide range of elevated environmental temperatures may be accompanied by different sets of trade-off regulations for plastic growth and development in plants [77].

*FLOWERING CONTROL LOCUS A* (*FCA*) acts in an autonomous pathway by repressing the floral repressor *FLOWERING LOCUS C* (*FLC*) and plays a significant role in the ambient temperature (thermosensory) pathway through the floral activator *FLOWERING LOCUS T* (*FT*) for flowering [78]. In addition to its role in floral induction, *FCA* was implicated in thermomorphogenesis through the epigenetic regulation of PIF4 activity because the *fca* mutant showed enhanced hypocotyl elongation compared to WT in response to the high-ambient-temperature treatment following normal-temperature conditions (23 °C for four days and then 28 °C for three days) [79]. Interestingly, the different experimental conditions, such as the continuous treatment of high ambient temperature (28 °C) from germination, led to different thermal responses in the *fca* mutant, such as severe seedling growth arrest by regulating the chlorophyll biosynthetic enzymes PROTOCHLOROPHYLLIDE OXIDOREDUCTASES (PORs), which control autotrophic development for plant growth [80]. Yang et al. [11] reported that the growth inhibition shown in the *fca* mutant was almost entirely rescued in the *gpa1 fca* but not in the *agb1 fca* double mutant under continuous high-ambient-temperature conditions. In addition, the reduced number of dividing cells in root meristems of the *fca* mutant was recovered in *gpa1 fca* but not *agb1 fca*. Interestingly, these rescued phenotypes shown in the *gpa1 fca* double mutant disappeared in the *gpa1 agb1 fca* triple mutant like the *fca* single mutant [11], suggesting that the epistatic relationship between *GPA1* and *AGB1* is essential for thermal adaptation. Therefore, these data suggest that G protein signaling plays a crucial role in the developmental trade-off associated with *FCA*-mediated thermomorphogenesis through cell proliferation.

### 2.6. Perspective on the Role of Heterotrimeric G Protein Signaling with Trade-Off Modulators

In plants and animals, SNF1/AMPK-related protein kinases (SnRKs) and TOR act as critical modulators in regulating the trade-off between growth and stress responses [4,81]. Plant SnRK1s, including *KIN10* and *KIN11* genes, are most closely related to yeast sucrose non-fermentable 1 (SNF1) and animal AMP-activated protein kinase (AMPK) [4,82]. They are activated in response to the changes in energy status caused by stressful conditions such as nutrient deprivation, darkness, inhibition of photosynthesis, and hypoxia [83]. An l-o-f *kin10 kin11* knockdown mutant showed a growth defect. In contrast, plants overexpressing *KIN10* showed enhanced starvation tolerance and delayed developmental senescence [83], suggesting that SnRK1s are important for growth and development and plant adaptation to stresses associated with energy homeostasis. TOR also plays an evolutionarily conserved role in sensing energy and nutrient status to regulate cell proliferation and overall plant growth [84]. Therefore, null mutations of *TOR* result in embryonic lethality [85]. Even inducible *tor* knockdown- or TOR kinase inhibitor-treated seedlings exhibit severe growth inhibition [86]. In addition to nutrient signals, TOR activity is negatively regulated in response to cold and osmotic stress signals, reflecting the stress status in terms of plastic plant growth and development [87,88]. Previous studies suggested that SnRKs and TOR mainly function in the trade-off for plants to adapt to environmental stress while ensuring maximum survival chances with minimal resources because SnRKs and TOR function antagonistically under normal and stress growth conditions [4,81].

In addition to the roles of plant G protein signaling in normal growth and development [61], phenotypic analyses using G protein component mutants have shown that G proteins are involved in the morphological changes and tolerant/sensitive effects associated with the trade-off against biotic and abiotic stresses in many plant species, including *Arabidopsis*, rice, and maize [8,10,58,89,90]. Nevertheless, there is no direct evidence of a correlation between G protein signaling and trade-off modulators such as SnRKs and TOR. The only possible link is the RGS1-mediated sugar sensing and response. In addition to HEXOKINASE1 (HXK1) and SnRK1/TOR acting as cytoplasmic glucose sensors, RGS1 senses glucose as a plasma membrane receptor [35,91]. Recognition of glucose as a ligand triggers RGS1 endocytosis, which negatively regulates GPA1 activity [40]. Therefore, activated G protein signaling and TOR appear to influence glucose-mediated growth by sensing the available nutrient resources, including glucose, in response to stress conditions (Figure 3).

In contrast to plants, mammalian G protein signaling was reported to act as a positive or negative upstream regulator of mTORC1 by phosphorylating the TOR and Raptor components of the mTORC1 complex. These phosphorylations were mediated by the GPCR-mediated protein kinase A (PKA) pathway [92]. Although plants have a deficiency in GPCR-mediated cAMP and PKA signaling, this does not exclude the evolutionary scenario of the functional link between G protein signaling and trade-off modulators. Furthermore, G protein signaling is involved in regulating the life–death decision of cells in *Arabidopsis*, maize, and rice [9,63,93,94], similar to TOR [14], suggesting the relevance of G protein signaling in the trade-off regulation through crosstalk with nutrient sensing and sugar signaling.

## 3. Conclusions

In contrast to the molecular processes mediated by GPCR activation in animals, plant G protein signaling was recently reported to be activated by the G protein regulator 7TM-RGS1 or RLKs, which are relatively abundant in plants. Although the details of the signaling process are different due to the composition and corresponding receptors, the sensing and response of G protein signaling to environmental stimuli, including hormonal, olfactory, biotic and abiotic stress, and nutrient signals, is likely to have evolved in a similar manner in animals and plants in terms of function [15,95]. The relationship between G protein signaling and important trade-off modulators, such as SnRKs and TOR, that sense nutrient and energy status is known in animals but is currently unclear in plants (Figure 3). Therefore, further studies will examine whether plant G protein signaling is directly or indirectly associated with the trade-off modulators. These efforts, through a clearer understanding of the trade-off regulation mechanisms, may provide new strategies for designing and breeding stress-tolerant crops that can reset energy allocation to avoid the penalty of growth inhibition.

## Figures and Tables

**Figure 1 plants-13-03239-f001:**
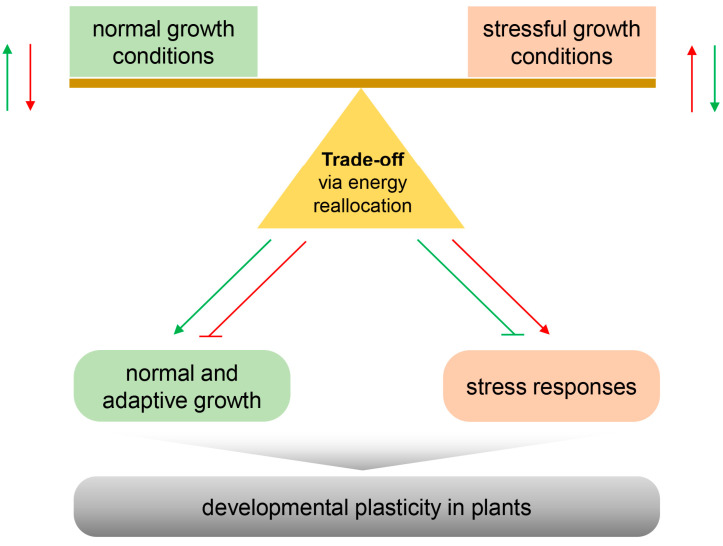
Hypothetical model of the relationship between trade-off and developmental plasticity. As sessile organisms, plants must respond to normal and stressful growth conditions. In general, plants under adverse stress show plastic deformation in growth and development through the underlying mechanism of trade-off between growth and stress responses, including changes in ROS, MAPKs, and gene expression. The green and red lines indicate the corresponding behavior under normal and stressful growth conditions, respectively.

**Figure 2 plants-13-03239-f002:**
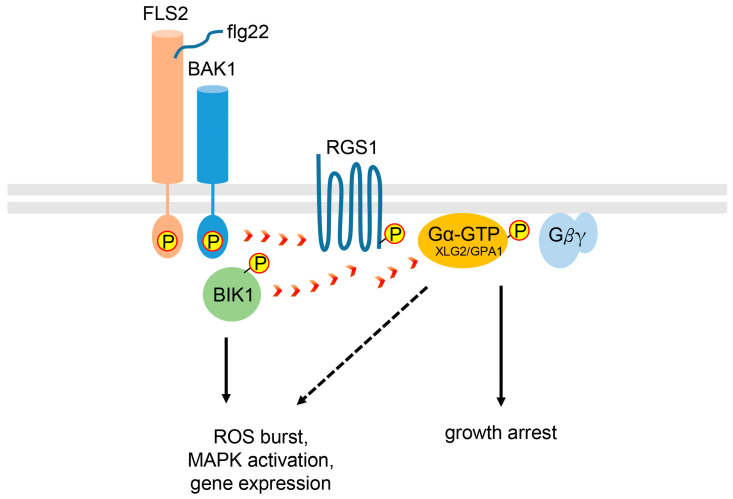
Schematic diagram of plant immune responses via heterotrimeric G protein signaling. Upon perception of flg22 signals, immune complexes, including receptors, cytoplasmic components, RGS1, and inactive G protein trimers, are dissociated. Activated kinases such as receptor BAK1 and cytoplasmic BIK1 phosphorylate 7TM-RGS1 and the Gα subunit to activate them (red arrowheads). A phosphorylated GTPase RGS1 is dissociated from the Gα subunit by endocytosis and the self-activating Gα subunit is activated by spontaneous exchange of GDP to GTP. Flg22 induces ROS burst, MAPK activation, and immune gene expression as an early response and growth arrest is likely to be mediated mainly by G protein signaling as a late response (solid arrows). Based on current data, G protein signaling is partly involved in the immune response for adaptation in response to biotic stress (dotted arrow).

**Figure 3 plants-13-03239-f003:**
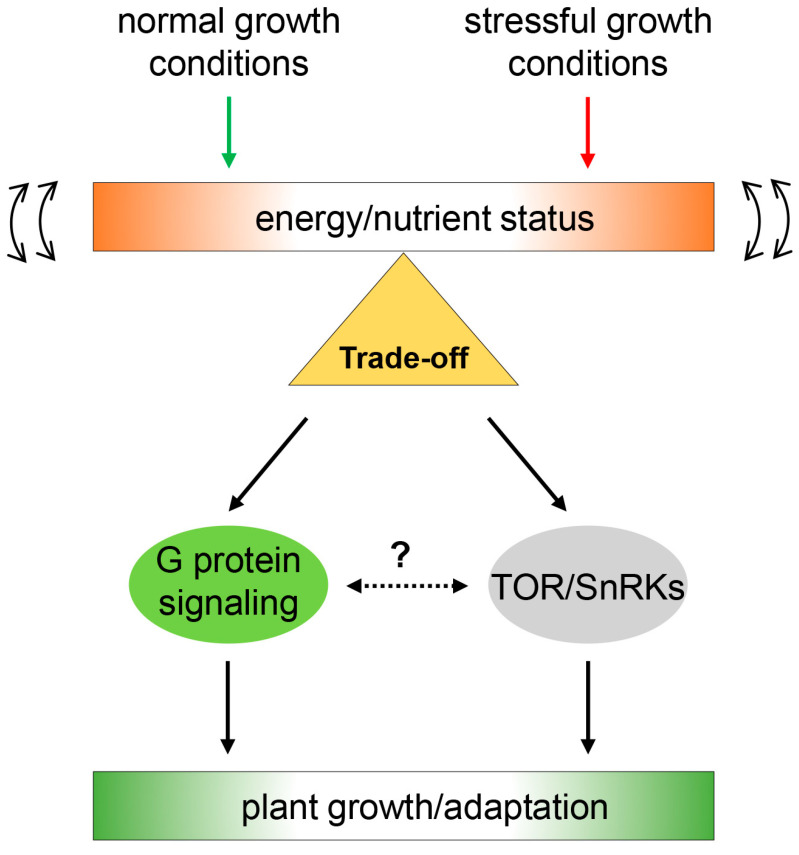
Hypothetical model of G protein signaling in the trade-off regulation. Energy status controlled by environmental conditions affects downstream signaling modules. Plant G protein signaling is involved in the regulation of plastic growth through RGS1, a putative glucose receptor. TOR and SnRKs as energy/nutrient sensors reciprocally act as trade-off modulators for plant growth and adaptation to environmental conditions. Open arrowheads indicate fluctuating energy and nutrient status controlled by normal and stressful growth conditions. Arrows indicate the involvement of downstream regulatory pathways for plant growth and adaptation to different environmental conditions. Dotted arrows indicate unidentified links between G-protein signaling and trade-off modulators.

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
