# Peer review of "Trade-Off Regulation in Plant Growth and Stress Responses Through the Role of Heterotrimeric G Protein Signaling"

_plants, 2024, doi:10.3390/plants13223239_

Round 1

Reviewer 1 Report

Comments and Suggestions for Authors

The review article written by Horim provides a comprehensive and in-depth discussion of the research on heterotrimeric G protein signaling in plants, especially their roles in shaping the trade-off between growth and stress responses. The review features a clear structure and coherent logic, providing readers with abundant background information and research foundations. In addition, the article excels in emphasizing the complexity of heterotrimeric G protein signaling and suggesting directions for future research. Overall, this article is a review with high academic value and is of great significance for promoting research and development in related fields. However, I have one concern on the review.

My concern:

The introduction provides a clear overview of the differences between plants and animals in terms of their ability to adapt to environmental conditions. The emphasis on plant developmental plasticity under stress conditions is well placed. However, it could be strengthened by including more recent findings or data to support the trade-off mechanism between growth and stress responses.

Author Response

Dear Editors and Reviewers,

I would like to submit our revised manuscript, entitled “Trade-Off Regulation in Plant Growth and Stress Responses Through the Role of Heterotrimeric G Protein Signaling” by Lee, for consideration of publication in Plants.

         Most of all, I really appreciate the thoughtful comments and suggestions of the editors and reviewers. I have carefully considered them and followed their suggestions to address their concerns and improve the manuscript. Detailed responses to the reviewers' comments are provided with each comment. In the revised manuscript, all editions have been tracked using the “Track Changes” function.

Response to Reviewer 1 Comments

The review article written by Horim provides a comprehensive and in-depth discussion of the research on heterotrimeric G protein signaling in plants, especially their roles in shaping the trade-off between growth and stress responses. The review features a clear structure and coherent logic, providing readers with abundant background information and research foundations. In addition, the article excels in emphasizing the complexity of heterotrimeric G protein signaling and suggesting directions for future research. Overall, this article is a review with high academic value and is of great significance for promoting research and development in related fields. However, I have one concern on the review.

My concern:

The introduction provides a clear overview of the differences between plants and animals in terms of their ability to adapt to environmental conditions. The emphasis on plant developmental plasticity under stress conditions is well placed. However, it could be strengthened by including more recent findings or data to support the trade-off mechanism between growth and stress responses.

Response: Thank you for the reviewers’ comments! As suggested, I updated the references with more recent work in the Introduction (1, 2, 3, 5, 10, 12, 13) and Conclusion (96) sections.

Response to Reviewer 2 Comments

Dear author,

I found the content of this paper intriguing. However, there are some points that should be addressed during the revision of this paper.

Comments 1- Please use the updated references to improve the introduction and discussion sections.

Response 1: Thank you for the reviewers’ comments! As suggested, I updated the references with more recent work in the Introduction (1, 2, 3, 5, 10, 12, 13) and Conclusion (96) sections.

Comments 2- Please add more tables and images to the content of your review.

Response 2: Thank you for the reviewers’ comments! As suggested, I added new Figures 2 and 3 to improve the manuscript.

Comments 3- The structural interaction of G proteins regarding the discussed topics in this paper should also be stated.

Response 3: Thank you for the reviewers’ comments! In this manuscript, I discussed the role of G protein signaling in plant immunity and thermomorphogenesis. Therefore, I mentioned the structural interactions of G protein subunits with immunity components in section 2.3. Of course, I also mentioned the structural interactions of G protein subunits with RGS1 for the basic molecular processes in section 2.2. However, as the role of G protein signaling in thermomorphogenesis has only recently emerged through the efforts of our group (Yang et al., 2023), there is little accumulated research to showing structural interactions.

Comments 4- How G protein signaling may affect the differentiation and growth of plant under harsh environmental stress?

Response 4: Thank you for the reviewers’ comments! Before responding to the reviewer’s comments, I would like to mention that the role of G protein signaling that I explain in this manuscript is likely to be involved in developmental plasticity based on the regulation of the trade-off between growth and stress responses. Therefore, I believe that the processes of G protein signaling associated with developmental plasticity and trade-off may be available under appropriate stress conditions. If plants are exposed to harsh environmental stress and the stress conditions are above the levels available for developmental plasticity and trade-off to work, plants would die under harsh environmental stress. Based on the current data, the role of G protein signaling appears to be involved in growth regulation under stress conditions available for developmental plasticity through trade-off regulation.

Comments 5- Please clearly mention that how many scientific databases were screened to find the relevant research or review papers regarding the discussed topic. Please briefly explain the search methodology conducted for writing this review paper.

Response 5: Thank you for the reviewers’ comments! When designing and writing this review, I tried to focus on the trade-off regulation of G protein signaling in plant immunity and thermomorphogenesis. Based on this context, I tried to find papers mainly using Web of Science. And I further searched for more related literature using the references in the papers selected from Web of Science.

Comments 6- Please use VosViewer to manage the academic literature related to the topic of this paper. There are many other papers in this line that you can use to improve your text's content.

Response 6: Thank you for the reviewers’ comments! When I tried to use the VosViewer program suggested by the reviewer, I realized that I need more time to use this program to get more meaningful information. Although I could not use this program to improve the manuscript in this time, we will use this program in other studies. Once again, thank you for letting us know about the useful program for managing the many papers for the manuscript.

Comments 7- Please add DOI identifiers to all cited literature.

Response 7: Thank you for the reviewer’s comments! I added the DOI information to each reference.

Comments 8- The conclusion section is too long. Please summarize the concluding remarks in five or six lines.

Response 8: Thank you for the reviewers’ comments! The first paragraph in the Conclusion section explained the difficulties of genetic engineering due to the trade-off regulation. So, I originally wrote the first paragraph to emphasize the last sentence, “…to avoid the penalty of growth inhibition”. However, since the first paragraph is almost independent of the second paragraph, I deleted the first paragraph of the previous Conclusion section.

Comments 9- Please summarize the content discussed in section 2.6 into a unique figure.

Response 9: Thank you for the reviewers’ comments! I added the Figure 3 for section 2.6.

Comments 10- There are different signaling pathways connected to G proteins that were not discussed in this paper. Please re-search the literature again and try to find the relevant papers.

Response 10: Thank you for the reviewers’ comments! As the reviewer mentioned, G proteins signaling is known to be involved in many different signaling pathways in addition to immunity and thermomorphogenesis mentioned in this manuscript. When preparing this manuscript, I tried to make the manuscript different from previously reported review papers that give general views of the roles of G protein signaling in plants. Therefore, in this manuscript I focused on the role of G protein signaling in immunity and thermomorphogenesis. Inevitably, to mention G protein signaling in plants, I followed the similar context in sections 2.1 and 2.2, which show the composition and molecular processes of heterotrimeric G proteins in plants. In section 2.3, although the role of G protein signaling in plant immunity is also frequently mentioned in other reviews, I provided the updated, controversial and recent advances of the conventional Gα subunit in flg22-mediated immunity in addition to the atypical Gα subunit, XLGs. In addition, the viewpoint of the role of G protein signaling in thermomorphogenesis was not reported in the previous review. I kindly ask the reviewer to understand this different scheme of the manuscript.

Comments 11- Although plant cells have most of the core elements found in animal G signaling, differences in network architecture and intrinsic properties of plant G protein elements make G signaling in plant cells distinct from the animal paradigm. In contrast to animal G proteins, plant G proteins are self-activating, and therefore regulation of G activation in plants occurs at the deactivation step. Please critically discuss the differences between animal and G proteins and their connected signaling pathways. Use such papers for this case: doi: 10.1146/annurev-arplant-050213-040133

Response 11: Thank you for the reviewers’ comments! As the reviewer mentioned, I agree with the difference in G protein signaling between plant and animal system. That is why I mentioned the different compositions in section 2.1 and the different molecular processes, including the self-activating Gα subunit, in section 2.2. Since I wanted to suggest the possible correlation of plant G protein signaling with trade-off modulators, such as TOR and SnRKs, I think this explanation probably makes the difference between plant and animal systems seem small. So, I removed the “slightly” in the Conclusion section (line 363) and edited sentences (lines 363-368) to make this clear.

Reviewer 2 Report

Comments and Suggestions for Authors

Dear author,

I found the content of this paper intriguing. However, there are some points that should be addressed during the revision of this paper.

1- Please use the updated references to improve the introduction and discussion sections.

2- Please add more tables and images to the content of your review.

3- The structural interaction of G proteins regarding the discussed topics in this paper should also be stated.

4- How G protein signaling may affect the differentiation and growth of plant under harsh environmental stress?

5- Please clearly mention that how many scientific databases were screened to find the relevant research or review papers regarding the discussed topic. Please briefly explain the search methodology conducted for writing this review paper.

6- Please use VosViewer to manage the academic literature related to the topic of this paper. There are many other papers in this line that you can use to improve your text's content.

7- Please add DOI identifiers to all cited literature.

8- The conclusion section is too long. Please summarize the concluding remarks in five or six lines.

9- Please summarize the content discussed in section 2.6 into a unique figure.

10- There are different signaling pathways connected to G proteins that were not discussed in this paper. Please re-search the literature again and try to find the relevant papers.

11- Although plant cells have most of the core elements found in animal G signaling, differences in network architecture and intrinsic properties of plant G protein elements make G signaling in plant cells distinct from the animal paradigm. In contrast to animal G proteins, plant G proteins are self-activating, and therefore regulation of G activation in plants occurs at the deactivation step. Please critically discuss the differences between animal and G proteins and their connected signaling pathways. Use such papers for this case: doi: 10.1146/annurev-arplant-050213-040133

Comments on the Quality of English Language

Please expand your discussion

Author Response

(The authors gave the same response as above.)

Round 2

Reviewer 2 Report

Comments and Suggestions for Authors

accept